# Equivalence Trial of the Non-Bismuth 10-Day Concomitant and 14-Day Hybrid Therapies for *Helicobacter pylori* Eradication in High Clarithromycin Resistance Areas

**DOI:** 10.3390/antibiotics13030280

**Published:** 2024-03-20

**Authors:** Sotirios D. Georgopoulos, Elias Xirouchakis, Christos Liatsos, Pericles Apostolopoulos, Panagiotis Kasapidis, Beatriz Martinez-Gonzalez, Fotini Laoudi, Maria Stoupaki, Georgios Axiaris, Dionysios Sgouras, Andreas Mentis, Spyridon Michopoulos

**Affiliations:** 1GI and Hepatology Department, Athens Medical, Paleo Faliron Hospital, 17562 Athens, Greece; elmoxir@yahoo.gr (E.X.); f.laoudi@yahoo.gr (F.L.); 2Gastroenterology Department, 401 General Military Hospital of Athens, 17562 Athens, Greece; cliatsos@yahoo.com; 3Gastroenterology Department, NIMTS Hospital, 11521 Athens, Greece; periclesapo@yahoo.com; 4Gastrenterology Department, Central Clinic of Athens, 10680 Athens, Greece; kasapendo@yahoo.gr; 5Laboratory of Medical Microbiology, Hellenic Pasteur Institute, 11521 Athens, Greece; bmartinez@pasteur.gr (B.M.-G.); sgouras@pasteur.gr (D.S.); mentis@pasteur.gr (A.M.); 6Gastroenterology Department, Alexandra General Hospital, 11528 Athens, Greece; maria_stoupaki@hotmail.com (M.S.); axiarisgeorge@gmail.com (G.A.); michosp5@gmail.com (S.M.)

**Keywords:** *H. pylori* eradication treatment, first line, hybrid treatment, concomitant treatment

## Abstract

**Background and aim**: We conducted an equivalence trial of quadruple non-bismuth “concomitant” and “hybrid” regimens for *H. pylori* eradication in a high clarithromycin resistance area. **Methods**: There were 321 treatment-naïve *H. pylori*-positive individuals in this multicenter clinical trial randomized to either the hybrid (esomeprazole 40 mg/bid, amoxicillin 1 g/bid for 7 days, then 7 days esomeprazole 40 mg/bid, amoxicillin 1 g/bid, clarithromycin 500 mg/bid, and metronidazole 500 mg/bid) or the concomitant regimen (all medications given concurrently bid for 10 days). Eradication was tested using histology and/or a 13C-urea breath test. **Results**: The concomitant regimen had 161 patients (90F/71M, mean 54.5 years, 26.7% smokers, 30.4% ulcer) and the hybrid regimen had 160 (80F/80M, mean 52.8 years, 35.6% smokers, 31.2% ulcer). The regimens were equivalent, by intention to treat 85% and 81.8%, (*p* = 0.5), and per protocol analysis 91.8% and 87.8%, (*p* = 0.3), respectively. The eradication rate by resistance, between concomitant and hybrid regimens, was in susceptible strains (97% and 97%, *p* = 0.6), clarithromycin single-resistant strains (86% and 90%, *p* = 0.9), metronidazole single-resistant strains (96% and 81%, *p* = 0.1), and dual-resistant strains (70% and 53%, *p* = 0.5). The side effects were comparable, except for diarrhea being more frequent in the concomitant regimen. **Conclusions**: A 14-day hybrid regimen is equivalent to a 10-day concomitant regimen currently used in high clarithromycin and metronidazole resistance areas. Both regimens are well tolerated and safe.

## 1. Introduction

The global human pathogen *Helicobacter pylori* (*H. pylori*) is linked to the pathophysiology of stomach cancer and peptic ulcers [1]. According to recent standards, *H. pylori* should be eliminated whenever it is discovered [2]. Selecting the right medications is crucial for a successful eradication [3,4].

Increasing clarithromycin resistance reduces the efficacy of standard triple therapy in many parts of the world [5,6,7]. Recent guidelines propose bismuth quadruple therapy as a rational approach for an empirical first-line *H. pylori* eradication treatment because it is a clarithromycin-free regimen [2]. However, many countries lack bismuth salts and/or tetracycline, and, thus, non-bismuth quadruple regimens with clarithromycin, metronidazole, and amoxicillin have proved to be the most effective alternative [8,9,10]. First-line tailored treatment (after susceptibility testing) has also been recommended, according to the rules of antimicrobial stewardship, but its generalized use in clinical practice remains to be established [2,6,11].

Among non-bismuth quadruple therapies, the concomitant regimen (proton pump inhibitor, amoxicillin, clarithromycin, and metronidazole for 10 days) seems to be superior in several countries including Greece [9,12,13]. Consequently, concomitant therapy of at least 10 days has been proposed as the preferred first-line regimen for *H. pylori* eradication by many national and international consensus [2,14,15,16,17,18]. In Greece, a 10-day concomitant regimen is highly effective, and as equally effective as the 14-day hybrid regimen [14,19,20]. A hybrid regimen that includes a proton pump inhibitor and amoxicillin for 10 to 14 days with the addition of clarithromycin and metronidazole during the last 5 or 7 days has been shown to be highly effective in various studies, mainly from Asian countries, but has been poorly studied in European populations [21,22]. Recently, the 14-day hybrid regimen reached, in a per protocol (PP) analysis, a >90% eradication rate [23]. Additionally, the 14-day hybrid therapy seems to be superior to 10 days in high clarithromycin resistance areas [24]. Published comparative studies between hybrid and concomitant therapy concern regimens of equal duration [22,25,26] and reported efficacy in both antibiotic-susceptible and antibiotic-resistant bacterial strains are scarce [23,27,28]. Thus, we conducted an equivalence trial for first-line *H. pylori* eradication between the 14-day hybrid and the 10-day concomitant regimen. We also aimed to evaluate the impact of antibiotic resistance to the abovementioned regimens and safety profiles.

## 2. Results

The concomitant scheme was assigned to 161 individuals and the hybrid to 160. Ultimately, 6 patients (3 in the hybrid group and 3 in the concomitant group) ended their treatment early because of serious adverse effects, and an additional 19 patients (10 in the concomitant and 9 in the hybrid group) were lost to follow-up. All 321 patients were included in the intention to treat (ITT) analysis. In the modified intention to treat (mITT) analysis, we included 302 patients who returned for the appropriate eradication testing. Finally, 296 patients (148 in the concomitant and 148 in the hybrid group) who completed the treatment protocol and returned for re-evaluation were included in the per protocol (PP) analysis. Before pooling the data of all five participating centers for analysis, we performed a stepwise logistic regression analysis between centers for both ITT and PP and found no significant differences in eradication rates for both treatments, (OR 0.85, 95% CI 0.6–1.1, *p* = 0.2), neither for concomitant (OR 0.79, 95% CI 0.5–1.1, *p* = 0.2), nor for hybrid (OR 0.9, 95% CI 0.6–1.3, *p* = 0.6). A flowchart of the included patients during the study is depicted in Figure 1.

Regarding the clinical and demographic features of the patients, there were no differences between the two therapy arms. Table 1 shows baseline clinical and demographic data.

### 2.1. H. pylori Eradication Rates

Pooled eradication rates in the concomitant and hybrid groups were equivalent, in the ITT [(85% vs. 81.8%, Difference = 3.2% (95% CI −4 to 11), *p* = 0.5], mITT [(90.7% vs. 86.7%, Difference = 4% (95% CI −3.2 to 11.3), *p* = 0.3], and PP analyses [91.8% vs. 87.8%, Difference = 4% (95% CI −3 to 11), *p* = 0.3] (Table 2).

### 2.2. Adverse Events and Adherence to Treatment

While 113 out of 302 patients (37.4%) reported adverse events, most were mild or moderate in intensity (88.5%) and only 13 out of 113 patients (11.5%) experienced severe side effects. For three patients on each treatment arm—or 1.9% of the total—severe adverse effects like vomiting, diarrhea, candidiasis, and epigastric pain led to their early treatment termination. The concomitant and hybrid treatments, respectively, caused side effects in 57/151 and 56/151 patients (37.7% and 37.1%, *p* = 0.9); more in-detail mild side effects were reported from 34/151 and 29/151 patients (22.5% and 19.2%, *p* = 0.6), moderate from 37/151 and 36/151 (24.5% and 23.8%, *p* = 0.8), and severe from 7/151 and 6/151 (4.6% and 3.9%, *p* = 0.6). In total, we recorded 195 adverse reactions. Diarrhea and taste disturbances were the most frequent complaints reported by 19.2% and 16.5% of the cases for the concomitant and 10.5% and 18.5% for the hybrid arms, respectively. However, the only significant difference was for diarrhea (*p* = 0.04). All side effects disappeared when treatment was stopped or discontinued. Every patient with moderate to severe diarrhea tested negative for *C. difficile*. With both regimens (98.8%, 95% CI 97–99% with concomitant vs. 99.6%, 95% CI 99–100% with hybrid), adherence to the treatment was excellent and comparable.

### 2.3. Effect of Antibiotic Resistance on H. pylori Eradication Rates

Of the 321 included patients to which bacterial cultures were sent, in 281 (87.5%), both cultures and antibiotic susceptibility tests were successfully completed. The numbers and percentages of positive cultures were evenly distributed among treatment groups (142/161, 88.1% in concomitant vs. 139/160, 86.8% in hybrid group, *p* = 0.8). In total, 78 (27.7%) and 97 (34.5%) out of 281 patients had primary resistance to metronidazole and clarithromycin, respectively. Of them, 33 (11.7%) individuals had dual-resistant strains (to metronidazole and clarithromycin). Levofloxacin primary resistance was low (8.1%) in our research population. No patient was noted to have triple resistance, and neither tetracycline nor amoxicillin resistance was detected. In terms of the prevalence of various patterns of antibiotic resistance, both groups were balanced. Table 3 shows the patterns of antibiotic resistance in each treatment arm.

Additionally, a sub-analysis of eradication rates based on antibiotic resistance profiles obtained from all patients included in the per protocol analysis (268/296) was carried out. With both concomitant and hybrid regimens, the eradication rates of dual-susceptible *H. pylori* strains (to metronidazole and clarithromycin) were excellent and comparable (97% vs. 97%, *p* = 0.6). A small not significant difference in favor of the hybrid regimen was recorded in clarithromycin single-resistant strains (86% vs. 90%, *p* = 0.9). In contrast, higher, but again, no significant eradication rates in favor of the concomitant regimen compared to the hybrid regimen were recorded in both metronidazole single-resistant (96% vs. 81%, *p* = 0.1) and dual-resistant (to clarithromycin and metronidazole) strains of *H. pylori* (70% vs. 53%, *p* = 0.5).

In order to further assess if the above findings depend on the level of resistance, we separated our patients who harbored resistant strains to high or low MIC values as previously described [29]. Our results, even if not statistically significant (due to type 2 error), show a tendency for a decreased efficacy of the hybrid compared to the concomitant regimen in dual high-resistant strains (Table 4). As compared to dual sensitive strains, only dual high-resistance strains showed statistically significant lower eradication rates with both regimens (66% vs. 97%, *p* = 0.001 for concomitant and 46% vs. 97%, *p* = 0.0001 for hybrid) and metronidazole single high-resistance strains for the hybrid regimen (78.5% vs. 97%, *p* = 0.01) but it should be taken into account that the number of low-MIC strains was very low in our study (Table 4).

### 2.4. Multivariate Analyses for Factors Influencing H. pylori Eradication

In the sample of 268 patients whose culture and antibiotic susceptibility tests were included in the PP analysis, we conducted a stepwise multivariate logistic regression analysis. The presence of single clarithromycin resistance (OR: 0.15 95% CI 0.03–0.7, *p* = 0.01) and dual resistance to both clarithromycin and metronidazole (OR: 0.07 95% CI 0.01–0.3, *p* = 0.001) were the only independent factors that significantly influenced the outcome of concomitant therapy. In contrast, single metronidazole (OR: 0.27 95% CI 0.07–0.97, *p* = 0.04) and dual resistance (OR: 0.06 95% CI 0.01–0.2, *p* = 0.0001) were the only independent factors that significantly influenced the outcome of hybrid therapy.

## 3. Discussion

The present multicenter randomized controlled study has been designed to prove equivalence between two *H. pylori* eradication regimens that, according to recently published data [9,12,23], still stand well in our area which has a high level of clarithromycin (>20%) and metronidazole (>30%) resistance. Our results show that the 14-day hybrid therapy is equivalent to 10-day concomitant therapy. Both regimens have reached acceptable eradication rates, 91.8% for the concomitant, and 87.8% for the hybrid, in the PP analysis. Similar results for these regimens have been recorded in the Greek database of the *H. pylori* European registry (Hp-EuReg) [30]. Our multivariate logistic regression analysis based on demographic, endoscopic, and antibiotic susceptibility data showed that antibiotic resistance has a significant negative influence on both regimens. The eradication rates for dual-susceptible strains were 97% for both regimens. However, the concomitant (OR: 0.07 95% CI 0.01–0.3, *p* = 0.001) and hybrid (OR: 0.06 95% CI 0.01–0.2, *p* = 0.0001) regimens were less efficient in patients harboring dual-resistant strains to clarithromycin and metronidazole, being 70% and 53%, respectively, whereas the single resistance to clarithromycin reduced the concomitant efficiency to 86% (OR: 0.15 95% CI 0.03–0.7, *p* = 0.01) and the single resistance to metronidazole reduced the hybrid efficiency to 81% (OR: 0.27 95% CI 0.07–0.97, *p* = 0.04).

Empirical *H. pylori* therapies are the most prescribed and are still proposed as first-line options, in parallel to the antibiotic susceptibility-based therapies, by major consensus conferences [2,16]. Despite the fact that antibiotic stewardship is largely becoming important for *H. pylori* schemes, availability, cost, and convenience prevents its widespread implementation [2,31,32]. Moreover, recent meta-analyses show that most susceptibility-guided therapies fail to overcome the acceptable 90% eradication rate, or even do not perform better than the empiric quadruple (with or without bismuth) regimens [31,32,33]. When bismuth alone or in an all-in-one capsule with antibiotics is available, it is the preferred first choice for empirical therapy [2]. One study from Taiwan [34] in a moderate clarithromycin (16%) and not very high metronidazole (25%) resistance area compared the efficacy of a 14-day bismuth quadruple therapy to the 14-day hybrid regimen and found comparable eradication rates (96% and 95%, respectively). The bismuth therapy had significantly more adverse effects.

Hybrid therapy was suggested, aiming at the hypothetical benefit of reducing the amount of antibiotics needed and consequently the cost and adverse effects. In a pilot study from 2011, 14-day hybrid therapy showed an outstanding performance of 99% by PP analysis in a country with low clarithromycin resistance (7%) [21]. Recently, Chen et al. confirmed high eradication rates by the 14-day regimen (93% in ITT and 95.5% in PP analysis) in a large retrospective study from Taiwan [35]. Hybrid therapy has also been studied as a 10-day scheme. In 2014, Wu et al. compared the efficacy of 10- and 14-day hybrid therapies for *H. pylori* eradication in Taiwan, an area of medium clarithromycin resistance, showing no significantly different PP eradication rates of 93%, and 95%, respectively [36]. However, a study from Iran, an area with a high clarithromycin resistance rate, showed significantly lower eradication rates for the 10-day hybrid therapy: 83% compared to 92% for the 14-day hybrid, meaning that the 10-day hybrid may not be a viable option in high antibiotic resistance areas [24]. In our country, the 14-day hybrid therapy [23] has been reaching eradication rates of 90%. In fact, the results of the present study show that the hybrid regimen achieved an acceptable eradication rate of 87.8% in the per protocol analysis. The major cause for decreased effectiveness of the hybrid therapy was the presence of dual-resistant *H. pylori* strains.

The non-bismuth quadruple concomitant therapy continues to achieve eradication rates of over 90%, in areas with high clarithromycin and metronidazole resistances, as far as dual resistance is not over 15% [37]. Concomitant therapy is mostly used in our country with a 10-day regimen based on recent studies that have shown eradication rates are over 90% in per protocol analysis [9,12,19]. Additionally, the 10-day concomitant regimen can still reach a relatively good result over the dual-resistant strains [9,10], and also the total number of pills is not importantly different from the 14-day hybrid regimen (80 vs. 84). All European studies to date have compared the 14-day hybrid regimen to the optimized 14-day concomitant regimen. In the first RCT published by Molina-Infante et al. [27], eradication rates were not significantly different (PP: 92% for hybrid and 96% for concomitant). Results based on antibiotic susceptibility are reported from a small cohort. The second study by De Francesco et al. [38] had several arms for the treatment, no antibiotic susceptibility tests, and its results were both excellent in PP analysis (95%) for the 14-day concomitant and hybrid regimens. In the third study by Mestrovic et al. [39], results were similar, 96% and 95% in PP analysis for the two 14-day therapies. The 10-day concomitant therapy has been compared with the 10-day hybrid in two studies; the first from Spain [40] showed eradication rates of 90% and 94% and the second from Korea [41] (where clarithromycin and metronidazole resistance are above 20% and 30%, respectively) showed 89% for both regimens. More recently, a comparison of a 10-day concomitant to a 14-day hybrid regimen has been presented in a study from Iran [42], with even lower eradication rates of 83% and 89%, respectively. These are largely conflicting results, and the major drawback is the lack of specific susceptibility tests. Instead, our study is the first study comparing the 10-day concomitant regimen to the 14-day hybrid regimen which also includes a regression analysis based on antibiotic susceptibilities with the larger number of patients reported to date. We not only show equivalence between these two regimens but we also found that both are influenced by single resistances (concomitant by clarithromycin and hybrid by metronidazole single resistance). More importantly, both are influenced by dual resistance to both antibiotics, as expected from previous studies [9,23]. Especially for the hybrid regimen, which is more vulnerable to dual resistance (eradication of only 53%), this may be due to the short duration of metronidazole in the second phase of treatment. Finally, in line with a previous study presenting patients from the 10-day sequential eradication regimen [29], we found that high dual-resistant profiles, such as more than 8 mg/L for clarithromycin and 32 mg/L for metronidazole, significantly lower eradication rates to 66% for the concomitant regimen and 46% for hybrid. On the contrary, the low resistance profile has shown excellent eradication rates, an observation that probably needs further attention and possibly a new look towards more clinically relevant limits of resistance.

Empiric regimens for *H. pylori* have mostly been using low-cost antibiotics. Additionally, the cost of PPIs is also relatively low. In our study, the two regimens have proved to be equivalent regarding eradication rates, but still, a question of cost-effectiveness may arise before choosing which one to use. A cost-effectiveness study from our country published in 2020 which also included costs for endoscopy, histology, and UBT tests for eradication showed that the 10-day concomitant regimen with esomeprazole had the lowest cost-effectiveness analysis ratio (EUR 179.17), whereas the hybrid regimen was found to be slightly higher (EUR 187.42) [43].

In accordance with prior studies and meta-analyses [22,27,38,41], both regimens demonstrated equivalent safety profiles and were well tolerated. In concomitant and hybrid therapies, the percentage of patients experiencing severe adverse effects was 4.6% and 3.9%, respectively, and no deaths were reported. With the exception of diarrhea, which was reported in a significantly larger percentage of patients (19.2%) receiving concomitant therapy than in those (10.5%) undergoing hybrid therapy (*p* = 0.04), the rates and severity of individual side effects were similar between regimens. This observation, even though not unusual [9,12], may be due to the adverse impact of more antibiotics used for longer periods of time on the diversity of gut micro-flora. Since increasing diarrhea is often presented during concomitant therapy, a recent study proved that the use of a probiotic capsule as an add-on to the 10-day concomitant therapy can significantly reduce diarrhea [44].

We believe there are two limitations in the current study. First, even while susceptibility tests from our metropolitan region and the rest of the nation have shown results that are comparable [9,45], our patient sample may not be indicative of the entire Greek population in terms of antibiotic susceptibility patterns. Second is the lack of CYP measurement in order to evaluate the profile of metabolizers, but since a previous study on the Greek population showed a large number of ultrarapid metabolizers [46], we used a new-generation PPI in high dosage. Thus, we consider both treatment arms, namely concomitant and hybrid, as having an optimized PPI coverage.

## 4. Conclusions

This study shows that a 14-day hybrid regimen is equivalent to a 10-day concomitant regimen which is currently used in a high clarithromycin (>20%) and metronidazole (>30%) resistance area. Both treatments could be successfully used as a first-line *H. pylori* treatment in Greece although the concomitant regimen is the one that consistently overcomes the threshold of 90% in terms of eradication rate. According to our susceptibility data, we can conclude that mostly, dual resistance to clarithromycin and metronidazole and to a lesser extent, single resistance to either clarithromycin or metronidazole, are negatively influencing both regimens.

## 5. Methods

### 5.1. Study Design

The research was an open-label randomized trial in design. The study, which was registered at ClinicalTrials.gov under the number NCT03592069, was conducted in the Gastrointestinal (GI) Departments of five participating hospitals: Alexandra General Hospital, 401 General Military Hospital, NIMTS Hospital, Central Clinic, and Athens Medical Paleo Faliron Hospital. It took place between February 2018 and January 2021. All local ethics committees approved the study protocol, which complied with ICH guidelines for good hospital practice and the principles of the Helsinki Declaration. Prior to enrollment, all patients involved in the trial provided written, fully informed consent.

### 5.2. Role of the Funding Source

A grant from the Hellenic Society of Gastroenterology supported this research. The study’s funders were not involved in its design, data collection, analysis, interpretation, or paper writing. Every author had complete access to the study’s data and was involved in its review and approval.

### 5.3. Patient Recruitment

Individuals aged eighteen years or above who were referred for an upper gastrointestinal endoscopy suffering from dyspepsia or iron deficiency anemia and were found to have a positive rapid urease test for *H. pylori* infection were invited to sign an informed consent form and take part in the study. These patients never received treatment for *H. pylori* eradication and met all eligibility criteria which were as follows: not having serious co-morbidities, not having undergone gastric surgery, not having cancer or Zollinger–Ellison syndrome, not being pregnant or lactating, not having a known allergy, and not having used bismuth salts or antibiotics within the previous two weeks.

Three hundred thirty (330) eligible patients were selected. Nine patients refused to sign the informed consent and were excluded. The remaining 321 patients were included in the study. A careful medical history was obtained, and complete clinical examinations were performed as well as appropriate blood tests and supplementary examinations if indicated prior to inclusion into the study.

### 5.4. H. pylori Detection

Eligible patients who entered the study had to test positive for a rapid urease test (CLO-test) using one biopsy specimen from the antrum and corpus. Two more specimens (from the corpus and antrum) were sent with all positive samples to the Hellenic Pasteur Institute, a reference laboratory, for testing for antibiotic susceptibility and bacterial culture.

### 5.5. Randomization and Masking

Individuals were randomized, one to one, to either the concomitant or hybrid therapy groups. A computer-based randomization with an independent central assistant investigator who used a separate number for each patient, was put in a block size of four without stratification, sealed, and maintained in his office for the duration of the study, was carried out. The researchers were required to contact the research assistant for the assigned regimen after receiving informed consent from the patient. The participating investigators added all data into a computer database for analysis. Similarly to other *H. pylori* study designs, the trial was not blind to clinicians or patients [47].

### 5.6. Interventions

Patients were randomized to the 10-day concomitant or 14-day hybrid therapy groups. The concomitant regimen consists of 500 mg of metronidazole bid, 500 mg of clarithromycin bid, 500 mg of amoxicillin bid, and 40 mg of esomeprazole bid. The hybrid regimen contained 40 mg of esomeprazole and 1 g of amoxicillin for 14 days whereas for the final 7 days, 500 mg of clarithromycin and 500 mg of metronidazole were added. In all regimens, esomeprazole was administered half an hour before meals, and antibiotics right away following meals. A printed handout with instructions on how to correctly take their medications and better follow their treatment plan was given to the patients. During the research period, the use of antibiotics or other medications that could affect the outcome of treatment was not allowed. Four to six weeks following the end of antibiotic therapy, the effectiveness of the treatment was assessed using a 13C-urea breath test (UBT), conducted in accordance with the accepted European methodology [48]. Histological examination was the preferred diagnostic test for patients who needed a follow-up endoscopy.

### 5.7. Tolerability and Adherence

Immediately following the end of the eradication therapy and during the final re-evaluation appointment, a specific questionnaire was performed in order to analyze side effects within a structured clinical interview. The patients were asked to categorize each adverse event they encountered during the interview as “mild”, which refers to temporary and well tolerated; “moderate”, which refers to discomfort that partially interferes with regular everyday activities; or “severe”, which refers to significant interference with patients’ daily activities. Serious side effects that were deemed to be incapacitating or potentially fatal had to be reported to a regulatory body (National Organization of Medicines). All patients were given a pre-structured printed table with all dosages displayed. Patients were asked to mark each time a pill was taken and bring it back, along with any tablets that were not taken, so that the number of pills taken during the course of treatment could be determined. A patient’s nonadherence rate was less than 90% of the prescribed dosage.

### 5.8. Bacterial Culture and Antibiotic Susceptibility Testings

#### 5.8.1. Isolation and Antibiotic Susceptibility Testing

As previously published, *H. pylori* strains were isolated from stomach biopsies [9]. Individual colonies and culture sweeps were gathered, then frozen at −80 °C until needed. E-test strips (BioMerieux, Marcy l’Étoile, France) were used to test the antibiotic susceptibility of *H. pylori* in accordance with the manufacturer’s instructions on Mueller Hinton agar medium (CampyPak-Plus, Beckton Dickinson, Cockeysville, MD) supplemented with 10% horse blood. Between 48 and 72 h later, the results were read. According to EUCAST, the following clinical breakpoints for minimum inhibitory concentration (MIC) were used to determine *H. pylori* antibiotic resistance: amoxicillin (>0.125 mg/L), metronidazole (>8 mg/L), tetracycline (>1 mg/L), levofloxacin (>1 mg/L), and clarithromycin (>0.5 mg/L) [49].

#### 5.8.2. Sample Size Calculation

The purpose of the study was to determine whether the two eradication regimens—the 10-day concomitant and the 14-day hybrid—were equivalent. According to the findings of other studies, both regimens had eradication rates of over 85% in intention to treat and over 90% in per protocol analyses [21]. International Statistical Rules (FDA) state that two regimens are equivalent if there is a difference of no more than 15% between confidence intervals in their eradication rates. Utilizing the Monte Carlo simulation system (500 × 500 runs) and considering that in recent Greek studies eradication rates for concomitant and hybrid regimens range from 85 to 90% in intention to treat analysis [9,12,23], we determined a sample size of 150 patients in each treatment arm (with a 10% dropout rate) in order to achieve an 80% power in the trial.

#### 5.8.3. Statistical Analysis

To assess the outcomes, we used the statistical tool SPSS version 13 (SPSS, Chicago, IL, USA). To compare proportions, the chi-square test was applied. To compare continuous non-parametric data, the *t* test was employed. The mean ± 1 SD is used to represent all continuous variables, whereas absolute numbers and relative frequencies are used to represent categorical variables. In patients with a final treatment outcome to either concomitant or hybrid therapy and available culture-susceptibility testing, factors influencing *H. pylori* eradication were assessed using stepwise multivariate logistic regression analysis. The outcomes of *H. pylori* eradication for each regimen—concomitant and hybrid—were taken into account as the dependent variable. The independent variables included smoking, age over or under 50, gender, body mass index (BMI) of over or under 25 kg/m^2^, alcohol consumption of over or under 21 units/week for men and 14 for women, presence of ulcer or non-ulcer disease, and single metronidazole, single clarithromycin, or dual to both metronidazole and clarithromycin resistance. Any *p*-value below 0.05 was regarded as significant.

## Figures and Tables

**Figure 1 antibiotics-13-00280-f001:**
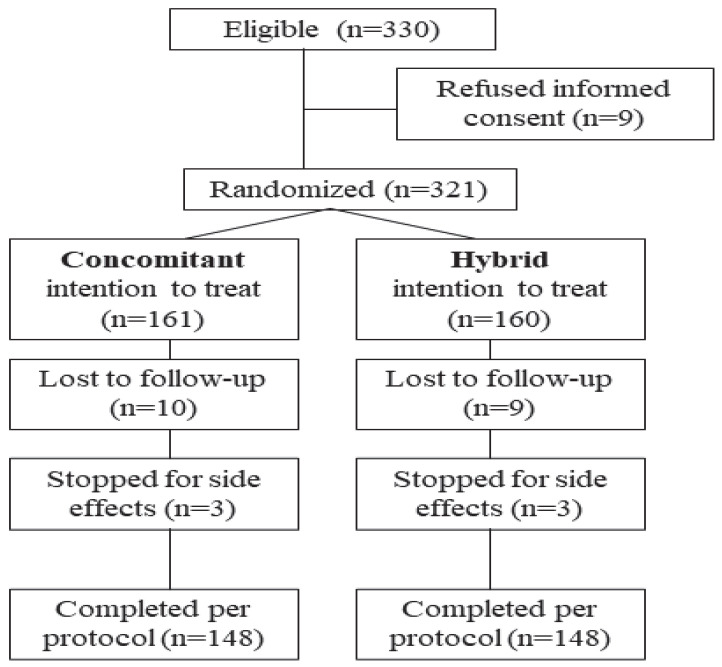
Flowchart of eligible patients during the study.

**Table 1 antibiotics-13-00280-t001:** Patient demographics and endoscopic features.

	Concomitant (n = 161)	Hybrid (n = 160)	*p*-Value
Gender M/F n (%)	71/90 (44–56)	80/80 (50–50)	0.3
Age (yrs) n (+/−SD)	54.5 (14.4)	52.8 (14.4)	0.3
BMI n (+/−SD)	26.4 (4.3)	26.2 (4)	0.6
Smokers n (%)	43 (26.7)	57 (35.6)	0.1
Alcohol use n (%)	25 (15.5)	23 (14.3)	0.8
*Endoscopic Findings*			0.4
NUD n (%)	112 (69.6)	110 (68.7)	
GUD n (%)	11 (6.8)	6 (3.7)	
DUD n (%)	38 (23.6)	44 (27.5)	

NUD = non-ulcer dyspepsia, GUD = gastric ulcer disease, DUD = duodenal ulcer disease.

**Table 2 antibiotics-13-00280-t002:** *H. pylori* eradication rates of the 10-day concomitant and 14-day hybrid treatment schemes.

Results	Concomitant	Hybrid	*p*-Value
Eradication rate ITT	137/161 (85%)	131/160 (81.8%)	*p* = 0.5
Eradication rate mITT	137/151 (90.7%)	131/151 (86.7%)	*p* = 0.3
Eradication rate PP	136/148 (91.8%)	130/148 (87.8%)	*p* = 0.3

ITT = intention to treat analysis, mITT = modified intention to treat analysis, PP = per protocol analysis.

**Table 3 antibiotics-13-00280-t003:** Antimicrobial resistance prevalence in patients receiving 14-day hybrid or 10-day concomitant therapy.

	Concomitant (n = 142)	Hybrid (n = 139)	*p*-Value
Amoxicillin resistance	0	0	
Tetracycline resistance	0	0	
Clarithromycin resistance	42 (29.6%)	36 (25.9%)	0.8
Metronidazole resistance	47 (33.1%)	50 (36%)	0.8
Dual resistance	18 (12.7%)	15 (10.8%)	0.7
Levofloxacin resistance	10 (7%)	13 (9.3%)	0.6
Triple resistance	0	0	

**Table 4 antibiotics-13-00280-t004:** *H. pylori* eradication rates of each regimen according to high or low MIC levels in the different patterns of resistance.

*H. pylori* Susceptibility Pattern and Level of Resistance	Concomitant	Hybrid	*p*-Value
**Dual-resistant strains (n)**	17	15	
Dual High Resistant	8/12 (66%)	6/13 (46%)	0.7
MET High and CLA Low Resistant	2/2 (100%)	0	NA
CLA High and MET Low Resistant	1/2 (50%)	2/2 (100%)	0.5
Dual Low Resistant	1/1 (100%)	0	NA
**CLA single-resistant strains (n)**	22	21	
CLA High Resistant	16/19 (86%)	11/13 (85%)	0.6
CLA Low Resistant	3/3 (100%)	8/8 (100%)	NA
**MET single-resistant strains (n)**	28	32	
MET High Resistant	19/20 (95%)	22/28 (78.5%)	0.2
MET Low Resistant	8/8 (100%)	4/4 (100%)	NA

NA = not applicable, MET = metronidazole, CLA = clarithromycin. High resistance has an MIC value of more than 8 to 256 mg/L for CLA and more than 32 to 256 mg/L for MET. Low resistance is more than 0.5 to 8 mg/L for CLA and more than 8 to 32 mg/L for MET.

## Data Availability

All data are available in electronic file kept in GI Department at Athens Medical Paleo Faliron Hospital (georgpap@ath.forthnet.gr).

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
