# Peer review of "Equivalence Trial of the Non-Bismuth 10-Day Concomitant and 14-Day Hybrid Therapies for Helicobacter pylori Eradication in High Clarithromycin Resistance Areas"

_antibiotics, 2024, doi:10.3390/antibiotics13030280_

Round 1

Reviewer 1 Report

Comments and Suggestions for Authors

The study includes both patients who have had COVID-19 and those who have not. During the pandemic, the consumption of antibiotics increased significantly. Could this have influenced the data on antibiotic resistance presented in this study?

Also, the genetic polymorphism for the MDR gene in the Greek population was not mentioned, and it is known that Pgp is important for the pharmacokinetics of drugs used to treat Helicobacter pylori infection.

Other manuscript parts are well-written and can be accepted without changes.

Author Response

Dear Reviewer,

Thank you for your comments. We set below our responces.

The study includes both patients who have had COVID-19 and those who have not. During the pandemic, the consumption of antibiotics increased significantly. Could this have influenced the data on antibiotic resistance presented in this study?

Response: In fact, no patient included in our study had an active COVID-19 infection because the access for gastroscopy at the study centers during the period of pandemic required a negative covid test. Past COVID-19 infections were not an exclusion criterion for the participation in the study, as the study started before the COVID-19 pandemic, but recent (during the previous month) antibiotic consumption was and it is stated in methods. In addition, the H. pylori resistance patterns to antibiotics used for its eradication in the present study (i.e. amoxycillin, clarithromycin and metronidazole) were similar to those recorded in our last randomized study before the COVID-19 pandemic (see ref. 9).

Also, the genetic polymorphism for the MDR gene in the Greek population was not mentioned, and it is known that Pgp is important for the pharmacokinetics of drugs used to treat Helicobacter pylori infection.

Response: The genetic polymorphism for the MDR gene and Pgp levels have not been included in our study so we cannot provide data about it. In Greece there is only one study reporting genotype frequencies of the C3435T polymorphism of MDR gene in a Greek population cohort which were: 26.1%, 47.6% and 26.1% for the wild type (CC), the heterozygous type (CT) and the homozygous mutant type (TT), respectively (ref.: Chatzigeorgiou N, Drakoulis N. Epitheorese Klinikes Farmakologias kai Farmakokinetikes 27(1):113-116 (2009)) but no study exploring the impact of these alleles or Pgp levels on H. pylori eradication rates. A previous small study from Croatia showed no impact of Pgp on H. pylori eradication when amoxicillin and metronidazole or clarithromycin were used (ref: Babic Z, World Journal of Gastroenterology 2005).

  1. pylori MDR usually emerges from mutations simultaneously driving resistance to one or more antibiotic classes that confers a cumulative MDR profile and this is what we are showing in our study. Additional MDR mechanisms may exist but with only limited or hypothetical clinical relevance to date. (ref.: Tshibangu-Kabamba, E., Yamaoka, Y. Helicobacter pyloriinfection and antibiotic resistance — from biology to clinical implications. Nat Rev Gastroenterol Hepatol18, 613–629 (2021) https://doi.org/10.1038/s41575-021-00449-x).

Reviewer 2 Report

Comments and Suggestions for Authors

Dear authors,

I have read with interest your manuscript titled " Equivalence trial of the non-bismuth 10-day concomitant and 14-day hybrid therapies for Helicobacter pylori eradication in high clarithromycin resistance area." I believe that this is a well-constructed manuscript with only a few minor issues that need to be addressed.

1. The term "H. pylori" should be italicized.

2. In the manuscript, it is mentioned that "In 281 out of 321 patients (87.5%), bacterial cultures and antibiotic susceptibility tests were successfully completed." However, it is unclear why some patients did not have bacterial cultures and antibiotic susceptibility tests performed. It would be helpful to provide further clarification on this point.

3. Table 4 shows that the number of patients in each group was not 161 and 160, as stated, because bacterial cultures and antibiotic susceptibility tests were completed in 281 patients. Therefore, the n for each group should be adjusted accordingly.

4. In the "Methods" section, under "H. pylori detection," it is mentioned that a reference laboratory was used for testing antibiotic susceptibility and bacterial culture. However, the word "culture" was missing the "re." It would be helpful to include this letter in the word "culture."

I hope that these comments are helpful in improving the manuscript.

Author Response

Dear Reviewer,

Thank you for your comments. Please find our point by point responce.

  1. The term "H. pylori" should be italicized.

Response: Done

  1. In the manuscript, it is mentioned that "In 281 out of 321 patients (87.5%), bacterial cultures and antibiotic susceptibility tests were successfully completed." However, it is unclear why some patients did not have bacterial cultures and antibiotic susceptibility tests performed. It would be helpful to provide further clarification on this point.

Response: It is relevant to note that the culture of H pylori is really challenging. H pylori is a fastidious pathogen that requires rapid transport in a special transport medium and must be transported for up to 24h at 4˚C to dedicated laboratories. In some cases, the quality of biopsy specimens such as low bacterial load in biopsy samples or the appearance of non-replicative non-culturable viable forms, the so-called coccoid forms, may impede the growth of bacteria. A recent study has reported that bacterial culture failed in at least 20% of infected patients when tested under optimal conditions, as occurred in the case of clinical trials (Zullo et al., Ann Gastroenterol 2022).

  1. Table 4 shows that the number of patients in each group was not 161 and 160, as stated, because bacterial cultures and antibiotic susceptibility tests were completed in 281 patients. Therefore, the n for each group should be adjusted accordingly.

Response: We adjusted the n for each group on Table 4 (now Table 3 as previous Table 3 was cancelled) as you suggested and removed relatively the first cell from this Table. We included this information in the text as follows:

"From the 321 included patients that bacterial cultures were sent in 281 (87,5%), both cultures and antibiotic susceptibility tests were successfully completed. Numbers and percentages of positive cultures were evenly distributed among treatment groups (142/161, 88.1% in concomitant vs. 139/160, 86.8% in hybrid group, p=0.8)."

  1. In the "Methods" section, under "H. pylori detection," it is mentioned that a reference laboratory was used for testing antibiotic susceptibility and bacterial culture. However, the word "culture" was missing the "re." It would be helpful to include this letter in the word "culture."

Response: Done

Reviewer 3 Report

Comments and Suggestions for Authors

This paper "Equivalence trial of the non-bismuth 10-days concomitant an 14 days hybrid therapy for Helicobacter pylori eradication on in high clarithomycin resistant area" is  interesting because it addresses a serious problem present in many countries, but, as reported by the Authors in lines 6-8 of page 6 and also in lines 6-7 at page 8 of the Discussion, it is similar to numerous other articles that deal with the same problem. Furthermore, the results are also compatible with others papers previously published although resistance to the antibiotics used is reported in this paper not previously described.

I would suggest lightening the article by removing some tables whose contents are described in the article, such as Table 3 and Table 5.

Author Response

Dear Reviewer,

Thank you for your suggestion. We have removed Tables 3 and 5. 

Accordingly in the new numerical order of tables Table 4 was named Table 3 and Table 6 was named Table 4.